# Electrochemically mediated disproportionation for selective formaldehyde upcycling in acid

Yun Song[1,2,9], Zhaohua Zhu [1,9], Tridip Das[3,9], Aarya D. Riasati[3], Jianjun Su[1,2], Weihua Guo [1,2], Yong Liu[1,2], Geng Li[1,2], Yinger Xin[1,2], Qiang Zhang [1,2], Mingming He[1,2], Ruixuan Wang[1,2], Rui Xue[1,2], Shenlong Zhao [4], Chuan Xia [5], Ben Zhong Tang [6,7], Marc Robert [8], Xin Wang [1], William A. Goddard III [3] & Ruquan Ye [1,2] ✉

Formaldehyde (FA) electrolysis is attractive for paired production of value-added chemicals. However, conventional electrolysis adopts alkaline electrolytes, which triggers FA self-disproportionation and severe feed loss. Here we introduce a sustainable and selective strategy for valorizing FA through electrochemically mediated disproportionation in acidic electrolytes. By leveraging a dual-electrode system consisting of a hydrophobic copper tetraminophthalocyanine layer (CuTAPc-layer) cathode and a Pt₂Ru bimetallic anode, we efficiently convert FA into methanol and formic acid at high Faradaic efficiencies of 93.2% and 91.3%, respectively. Compared with alkaline FA oxidation, which can lose up to 76% FA and complicate downstream separation, the acidic system suppresses side reactions to ensure high product purity. Mechanism studies reveal that the hydrophobic microenvironment of CuTAPc-layer suppresses hydrogen evolution, while the stronger oxophilicity of Pt₂Ru enhances FA activation and lowers the key deprotonation barrier for FA oxidation. The integrated device demonstrates application potential in polyoxymethylene upgrading, delivering 374.2 mA at 4 V with ~90% single-pass conversion, establishing a scalable and eco-friendly electrochemical pathway for chemical upcycling.

Plastics are versatile materials used across various industries, but their environmental impact remains a critical concern[1–10]. Global plastic production reached 413.8 million metric tons in 2023 and is projected to increase to 800 million metric tons by 2040[11,12]. However, only 9% of the world's plastic waste is recycled, while nearly 80% is either discarded or landfilled, severely polluting the natural environment[13]. Among commodity plastics, formaldehyde-containing plastics, particularly polyoxymethylene, stand out for their exceptional properties[14].

[1]Department of Chemistry and State Key Laboratory of Marine Environmental Health, City University of Hong Kong, Hong Kong, China. [2]City University of Hong Kong Shenzhen Research Institute, Shenzhen, China. [3]Materials and Process Simulation Center, California Institute of Technology, Pasadena, CA, USA. [4]CAS Key Laboratory of Nanosystem and Hierarchical Fabrication, CAS Center for Excellence in Nanoscience, National Center for Nanoscience and Technology, Beijing, China. [5]School of Materials and Energy, University of Electronic Science and Technology of China, Chengdu, China. [6]Guangdong Basic Research Center of Excellent for Aggregate Science, School of Science and Engineering, Shenzhen Institute of Molecular Aggregate Science and Engineering, The Chinese University of Hong Kong, Shenzhen (CUHK-Shenzhen), Shenzhen, China. [7]Department of Chemistry and the Hong Kong Branch of Chinese National Engineering Research Center for Tissue Restoration and Reconstruction, The Hong Kong University of Science and Technology, Hong Kong, China. [8]Sorbonne Université, Institut Parisien de Chimie Moléculaire (IPCM), CNRS, Institut Universitaire de France (IUF), Paris, France. [9]These authors contributed equally: Yun Song, Zhaohua Zhu, Tridip Das. ✉e-mail: ruquanye@cityu.edu.hk

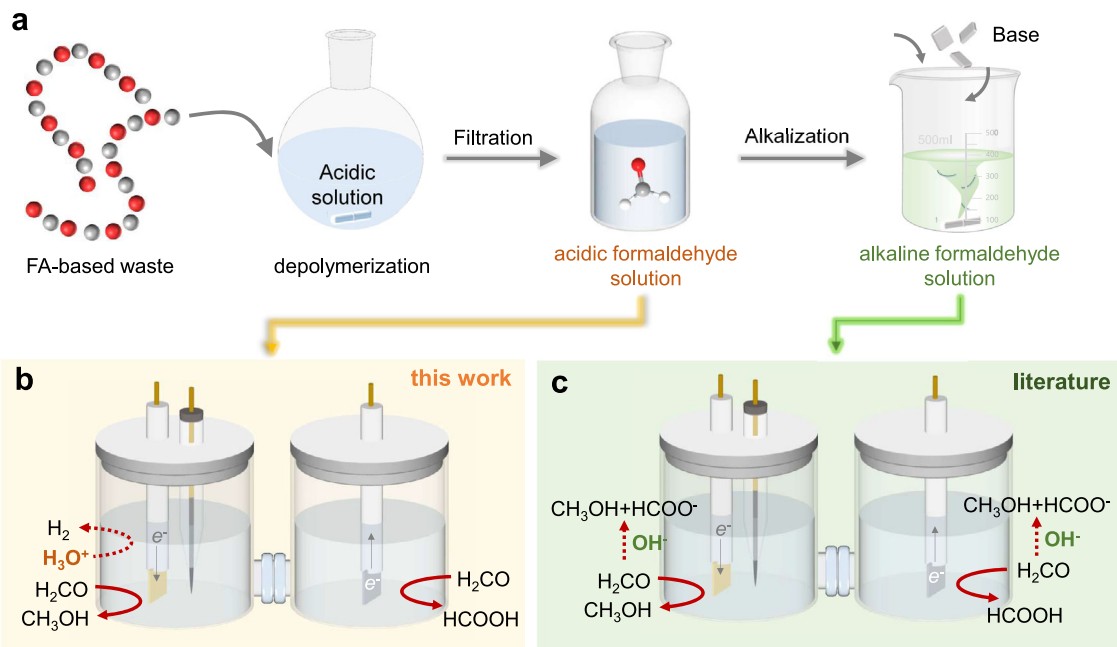

**Fig. 1 | Overview of the recycling of POM wastes. a** Depolymerization of formaldehyde-based waste into formaldehyde solution. The gray, red, and white spheres represent carbon, oxygen, and hydrogen atoms, respectively. Hydrogen is omitted in the polymer structure. Electrochemical formaldehyde conversions in acidic (**b**) and alkaline (**c**) media. Dashed arrows represent side reactions.

Polyoxymethylene (POM) possesses great flowability in injection molding and extrusion processes, facilitating the manufacture of highly precise and structurally intricate components[15]. The precision and mechanical strength of POM make it indispensable for automobiles, industrial machinery, and medical devices[16,17]. The global POM market has substantially grown in recent years. Its market size in 2024 is 5.02 billion USD, which is anticipated to reach 6.28 billion USD by 2028 at an annual growth rate of 5.7%[18]. However, this rising demand has led to mounting POM waste, necessitating improved recycling methods to mitigate environmental impact. Traditional disposal methods such as incineration[19], pyrolysis[20], mechanical recycling[21–23], and landfill[24–28] have significant drawbacks. For example, thermal methods such as incineration and pyrolysis produce carcinogenic gaseous FA as the main product, which requires stringent flue-gas treatment at high costs[29]. Mechanical recycling degrades the materials, resulting in inferior properties[30], and landfilling leads to long-term degradation processes, which contaminates soil and groundwater[27]. These methods fail to meet the growing demand for safe, sustainable, and environmental-friendly POM waste disposal.

Compared to conventional disposal methods, chemical transformation of polymers into value-added chemicals offers an attractive alternative for waste remediation[31–39]. There are two types of POM upcycling methods. One is based on the classic addition reaction between FA and alcohol to form dialkoxymethane[40,41]. The other is based on the FA redox reaction to form formic acid or methanol. However, the yield of redox process is typically poor. For example, Milstein developed a one-pot process of converting POM into methanol through acidolysis and Mn-catalyzed disproportion at a high temperature of 150 °C[42]. However, one-third of the FA was lost as $CO_2$, limiting the yield of methanol.

Electrochemically mediated conversion (EMC) reactions provide an emerging pathway for polymer remediation[43–46]. Compared to the conventional thermal process, EMC can be powered by renewable electricity at ambient conditions. In 2023, Moore et al. reported the EMC of POM in organic media, using hexafluoropropanol as both the solvent and proton donor in organic media[47]. They achieved effective depolymerization to formaldehyde, oxydimethanol, and 1,3,5-trioxane, thereby providing an electrochemical pathway towards sustainable plastic circular economy powered by renewable electricity. Since POM can be depolymerized to afford FA, electrochemical FA conversion (such as FA oxidation) has been proposed as an alternative half-reaction to the oxygen evolution reaction in water electrolysis[48–53]. However, the electrochemical aldehyde oxidation reaction (in which the aldehyde is oxidized to a carboxylate with spontaneous $H_2$ production) must be performed in alkaline solutions and relies on active $Cu^0$ sites. Under such conditions, these $Cu^0$ sites are readily oxidized to $Cu^+/Cu^{2+}$ species[54], compromising catalyst stability and performance. Additionally, alkaline electrolytes facilitate the spontaneous Cannizzaro disproportionation reaction, where two formaldehyde molecules react to produce one methanol and one formate (Fig. 1c). This side reaction reduces the FA conversion efficiency and complicates product separation and purification.

Given that POM is chemically stable in alkaline conditions, its hydrolysis is typically performed in acid. Thus, the electrolysis in alkaline conditions will inevitably increase the cost due to the additional alkalization step (Fig. 1a). The influence of electrolyte pH on operation cost has been increasingly recognized in related electrochemical systems[55,56]. For example, in alkaline $CO_2$ reduction flow cells, the application of alkaline electrolytes can inhibit the undesired hydrogen evolution reaction (HER), thereby increasing the selectivity of $CO_2$ reduction products. Nonetheless, alkaline conditions promote carbonate formation, resulting in substantial electrolyte consumption, which significantly decreases single-pass $CO_2$ utilization. Consequently, electrolyte regeneration can account for more than 50% of the overall operational cost, underscoring the significant economic disadvantages associated with alkaline electrolytes. Similarly, for POM conversion, we find that up to 76.1% FA is lost within 3 h due to FA disproportionation in 1 M KOH solution (Supplementary Note 1). This rapid reactant loss further elevates material costs for electrochemical FA conversion, highlighting the inherent economic drawbacks of employing alkaline conditions for POM upgrading. These challenges underscore the need for innovative approaches to enable selective and efficient POM upcycling while minimizing side reactions.

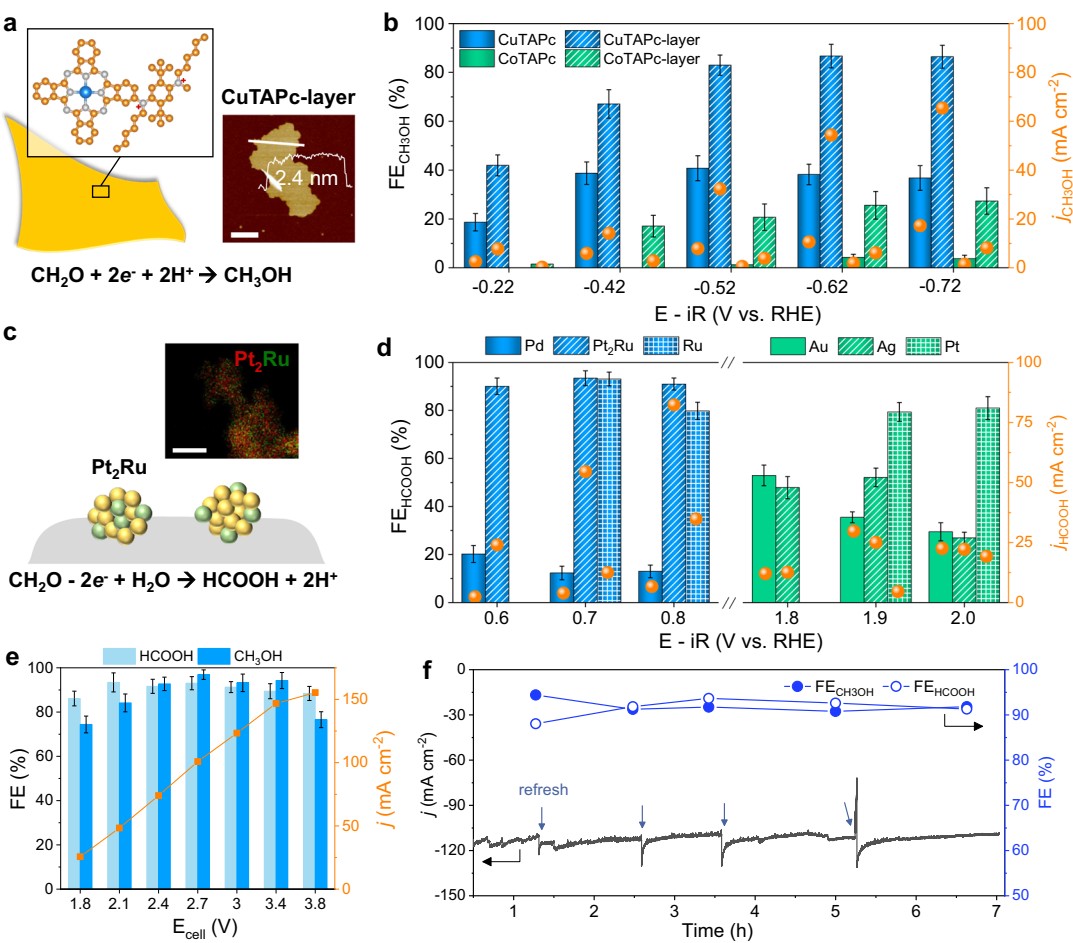

**Fig. 2 | Electrochemical performance evaluation of CuTAPc-layer and Pt$_2$Ru catalysts in acidic conditions. a** Structural illustration and AFM image of CuTAPc-layer, scale bar: 400 nm. The orange, blue, and white spheres represent carbon, cobalt, and nitrogen atoms, respectively. **b** FE$_{CH3OH}$ and $j_{CH3OH}$ of CoTAPc, CoTAPc-layer, CuTAPc, and CuTAPc-layer in an H-cell, catholyte: 1 M FA + 0.5 M K$_2$SO$_4$ + 0.05 M H$_2$SO$_4$, anolyte: 0.5 M H$_2$SO$_4$. **c** Structural illustration and HAADF-STEM mapping image of Pt$_2$Ru, scale bar: 25 nm. The green and yellow spheres represent platinum and ruthenium atoms, respectively. **d** FE$_{HCOOH}$ and $j_{HCOOH}$ of 20% Pd/C, Pt$_2$Ru alloy, Ru nanoparticles, Cu nanoparticles, 40% Au/C, Ag nanoparticles, and Pt foil in an H-cell, anolyte: 1 M FA + 0.5 M K$_2$SO$_4$ + 0.5 M H$_2$SO$_4$,

catholyte: 0.5 M H$_2$SO$_4$. **e** Catalytic performance of the paired CuTAPc-layer and Pt$_2$Ru catalysts as the cathode and anode in a combined flow cell, catholyte: 1 M FA + 0.5 M K$_2$SO$_4$ + 0.05 M H$_2$SO$_4$, anolyte: 1 M FA + 0.5 M K$_2$SO$_4$ + 0.5 M H$_2$SO$_4$. The electrocatalytic performance was measured without iR compensation. **f** Chronoamperometric stability of CuTAPc-layer/Pt$_2$Ru catalysts at a full cell voltage of 3 V, catholyte: 1 M FA + 0.5 M K$_2$SO$_4$ + 0.05 M H$_2$SO$_4$, anolyte: 1 M FA + 0.5 M K$_2$SO$_4$ + 0.5 M H$_2$SO$_4$. The error bars represent the standard deviation of three independent measurements. Source data for the figure are provided as a Source data file.

Electrochemical upcycling in acidic environments can eliminate the need of alkalization and prevent the Cannizzaro reaction. Despite these benefits, the electrochemical upcycling of POM waste in acidic solutions has been rarely studied. The challenges come from the detrimental HER at the cathode[57] and poor stability of catalysts at the anode in acidic electrolytes[58,59]. Herein, we report an acidic route for direct upgrading of POM into valuable chemical via two steps: (1) acidolysis of the POM to FA monomer and then (2) FA electrolysis in acidic conditions. By precise interface design and catalyst selection, we can efficiently convert FA into methanol at the cathode and formic acid at the anode, both with FEs of greater than 90%. We further achieve high single pass conversion efficiency (SPCE) of 86% and 89.5% for FA-to-CH$_3$OH and FA-to-HCOOH conversions, with a total current of 374.2 mA at a cell voltage of 4 V. Our technoeconomic analysis suggests that this POM valorization in acid provides a viable and sustainable pathway for waste management and resource recovery. In addition, our FA redox half-reactions in acid can be coupled with other reactions for paired electrolysis to improve energy efficiency and reduce carbon footprint.

## Results

### Screening of electrocatalysts for acidic formaldehyde reduction and oxidation

We start with the screening of electrocatalysts for FA redox reactions. For the FA reduction reaction, we focus on molecular catalysts, particularly metallophthalocyanines such as cobalt tetraminophthalocyanine (CoTAPc) and CuTAPc[60-64], through which formaldehyde is a possible intermediate. FA solution can be readily obtained from depolymerization of POM or paraformaldehyde. We first evaluated the activity in 1 M FA + 0.5 M K$_2$SO$_4$ electrolyte. The FA concentration is determined by ultraviolet-visible (UV-vis) absorption using the Hantzsch reaction method (Fig. S1), and the products are quantified using hydrogen nuclear magnetic resonance spectroscopy ($^1$H NMR) with dimethyl sulfoxide (DMSO) as the internal standard (Fig. S2). Figure 2b summarizes the maximum CH$_3$OH Faradaic efficiency (FE$_{CH3OH}$) and the corresponding CH$_3$OH partial current density ($j_{CH3OH}$) for different samples. Pristine CuTAPc and CoTAPc molecules show moderate/poor FA reduction activity. Due to the competing HER in acid, they attain a maximum FE$_{CH3OH}$ of 40.8 and 3.8%, respectively (Figs. S3 and S4).

To improve the FA reduction performance in acid, we leverage our former strategies in molecular engineering:

(1) the formation of ultrathin covalent organic nanosheet layers to improve molecular dispersion and electron transfer, and (2) the post-synthetic modification of layers to form hydrophobic interfaces to inhibit the side HER. Fig. S5 shows this scheme for converting CuTAPc into layered form, termed as CuTAPc-layer. Inductively coupled plasma optical emission spectrometry (ICP-OES) reveals that the copper contents of CuTAPc and CuTAPc-layer are 8.78 and 2.65%, respectively. The CuTAPc-layer shows a broad X-ray diffraction (XRD) peak at 21.2°, indicating its amorphous nature (Fig. S6a). Transmission electron microscopy (TEM; Fig. S6b) and atomic force microscopy (AFM; Fig. 2a) confirm its ultrathin structure, with a thickness of 2.4 nm. High-angle annular dark-field (HAADF) scanning transmission electron microscopy (STEM) element mapping images unravel a homogeneous distribution of Cu, N, and C elements within the framework (Fig. S6c). Fourier transform infrared (FTIR) spectroscopy and X-ray photoelectron spectroscopy (XPS) further provide insights into the chemical bonds. For FTIR (Fig. S7a), the disappearance of N-H stretching vibrations of the amino groups at 3318, 3199 cm$^{-1}$ and the appearance of a new C=N stretching vibration at 1655 cm$^{-1}$, C-N$^+$ bond at 973 cm$^{-1}$, and the alkyl group at 2800–3000 cm$^{-1}$, verify the successful preparation of alkyl-group-grafted cationic iminium species. This also agrees with the N 1$s$ spectrum, showing the emergence of iminium-N at 401.7 eV for CuTAPc-layer (Fig. S8a). However, due to the paramagnetic property of CuTAPc[65,66], the phthalocyanine carbon signal in the solid-state $^{13}$C NMR spectra is weak, and only the alkyl chain in the CuTAPc-layer is detected at 25 ppm (Fig. S7b).

The formation of layered structures changes the physical and chemical properties of molecules. The grafting of alkyl group endows CuTAPc-layer with hydrophobicity, increasing the water contact angle from 30.3° (CuTAPc) to 125.3° (Fig. S9). Zeta potential analysis verifies that CuTAPc-layer is positively charged, with a value of +37.8 mV (Fig. S10a, b). This electron-withdrawing effect also induces a blueshift of the Q band in the UV-vis spectra (Fig. S10c), consistent with the shift of Cu 2$p$ peak by 0.4 eV toward higher binding energy (Fig. S8b). This comprehensive characterization illustrates the successful formation of a hydrophobic ultrathin layer with high molecular dispersion.

As expected, the CuTAPc-layer achieves a maximum FE$_{CH3OH}$ of 86.7% at −0.62 V vs. reversible hydrogen electrode (RHE) and maintains a high methanol selectivity of >80% within the potential range of −0.52 to −0.72 V vs. RHE (Fig. 2b). Online gas chromatography analysis of the gaseous products reveals hydrogen as the primary byproduct. The hydrogen Faradaic efficiency (FE$_{H2}$) of CuTAPc-layer decreases with increasing applied voltage and reaches 11.3% at −0.57 V (Fig. S11). In contrast, CuTAPc (Fig. S3) maintains FE$_{H2}$ values above 55% over the range of −0.12 to −0.82 V. Similar inhibition of hydrogen evolution is observed for CoTAPc-layer, with FE$_{CH3OH}$ improving from 3.8% to 27.3% at −0.72 V vs. RHE. We also compare the acidic FA reduction performance of Cu and Cu$_2$O nanoparticles (Figs. S13 and S14), which are active catalysts for CO$_2$ reduction[67–70]. However, they both present an unstable current density in pH 1 electrolytes. Particularly, Cu shows a significant current density decay from 72 to 20.5 mA cm$^{-2}$ at −0.82 V vs. RHE, with FE$_{CH3OH}$ of only 20.5%. These results confirm the crucial role of hydrophobic interfaces in suppressing competing HER and enhancing methanol production efficiency for acidic FA reduction.

We next seek to identify electrocatalysts for acidic FA oxidation. We first compare the activity of a series of metals in acidic electrolytes, including Pd, Ru, Cu, Au, Ag, and Pt (Figs. S15–S22). Among these materials, Pt and Ru stand out with HCOOH Faradaic efficiency (FE$_{HCOOH}$) exceeding 70% (Fig. 2d). Pt generally exhibits a higher selectivity than Ru, but the working potential is much higher. To further optimize the performance, PtRu alloys with different Pt:Ru atomic ratios of 2:1, 4:1, 1:2, and 1:1 were synthesized (Methods). Take Pt$_2$Ru as an example. Pt$_2$Ru shows a spherical shape with an average

size of ~50 nm (Fig. S23). Elemental mapping suggests a homogeneous distribution of Pt and Ru throughout the alloyed catalyst (Fig. 2c). XRD analysis indicates the co-existence of Pt and Ru in the Pt$_2$Ru phase. The shifts of XPS Pt 4$f$ peak to higher binding energy and the Ru 3$d$ peak to lower binding energy suggest charge transfer within the alloy structure (Fig. S24). The formation of this Pt$_2$Ru alloy improves the FA oxidation (Figs. S25–S30). At an optimal composition of Pt$_2$Ru, the FE$_{HCOOH}$ and HCOOH partial current density ($j_{HCOOH}$) improves to 90.9% and 82.3 mA cm$^{-2}$ at 0.8 V vs. RHE.

To demonstrate the continuous conversion of FA, we use a flow cell with CuTAPc-layer and Pt$_2$Ru as the cathode and anode catalysts. Both anolyte and catholyte contain 1 M FA, but we control the pH to be 0 and 1 in anolyte and catholyte, respectively, so as to inhibit the crossover of OH$^-$ produced at the cathode. Current densities are measured at different full cell potentials. As shown in Fig. 2e, both the anode and cathode maintain high FEs over a wide cell potential range from 2.1 to 3.4 V. The integrated flow cell delivers a current density of 146.8 mA cm$^{-2}$ at a full cell voltage of 3.4 V, with FE$_{CH3OH}$ and FE$_{HCOOH}$ of 94.2 and 89.4%, respectively. This long-term stability test demonstrates that the integrated device system maintains a stable current density of approximately 110.6 mA cm$^{-2}$ at a cell voltage of 3 V over 7 h (Fig. 2f). During the long-time electrolysis, the full cell is refreshed by slowly flowing deionized water through the cathode chamber to dissolve deposited potassium salt on the surface of CuTAPc-layer. Electrolytes sampled at different times are analyzed (Fig. S31), and the FE$_{HCOOH}$ at the anode and FE$_{CH3OH}$ at cathode both remain above 88%. The catalysts after long-term electrolysis were collected and characterized. CuTAPc-layer maintains the nanosheet morphology after electrocatalytic reduction (Figs. S32 and S33). The combined results of XPS, FTIR, and UV-vis spectra further reveal that the composition and structure of CuTAPc-layer are well retained. Post-electrolysis characterization of Pt$_2$Ru also displays negligible changes in morphology, crystallinity, and electronic structure (Fig. S34). Therefore, the integrated CuTAPc-layer/Pt$_2$Ru full cell demonstrates high stability and efficiency in acidic electrolytes, making it promising for sustainable electrochemical upcycling of FA-based waste.

## Disproportionation reactions in neutral and alkaline environments

Prior studies on FA conversion have predominantly focused on alkaline environments[49,71], which inevitably induce disproportionation reactions and restrict selective transformation pathways. To explore catalytic performance and selectivity, we further conduct electrolysis using CuTAPc-layer and Pt$_2$Ru in both neutral and alkaline conditions (Figs. S35–S38). The reduction of FA in neutral electrolytes consumes protons, gradually generating an alkaline environment and subsequently triggering disproportionation reaction. The FE of methanol is calculated as below:

$$FE_{Cannizzaro+Reduction} = \frac{2 * 96485\,C/mol * n_{CH3OH,total}}{total\ charge} * 100\% \quad (1)$$

$$FE_{Reduction} = \frac{2 * 96485\,C/mol * (n_{CH3OH,total} - n_{HCOOH})}{total\ charge} * 100\% \quad (2)$$

As shown in Fig. 3a, the disproportionation reaction occurs even at a low current density of 16.8 mA cm$^{-2}$ at −0.4 V vs. RHE in 0.5 M K$_2$SO$_4$ and amplifies with higher overpotentials. Conversely, FA oxidation in 0.5 M K$_2$SO$_4$ generates protons, resulting in an acidic electrolyte where the disproportionation reaction is negligible. Figure 3c exhibits the $^1$H NMR spectra of the electrolytes from Fig. 3a, b, revealing the formation of disproportional products. Note that the signal of formic acid shifts across different electrolytes due to pH variations[72]. In 1 M KOH, the amount of methanol produced via disproportionation reaction increases significantly, reaching 3–4 times

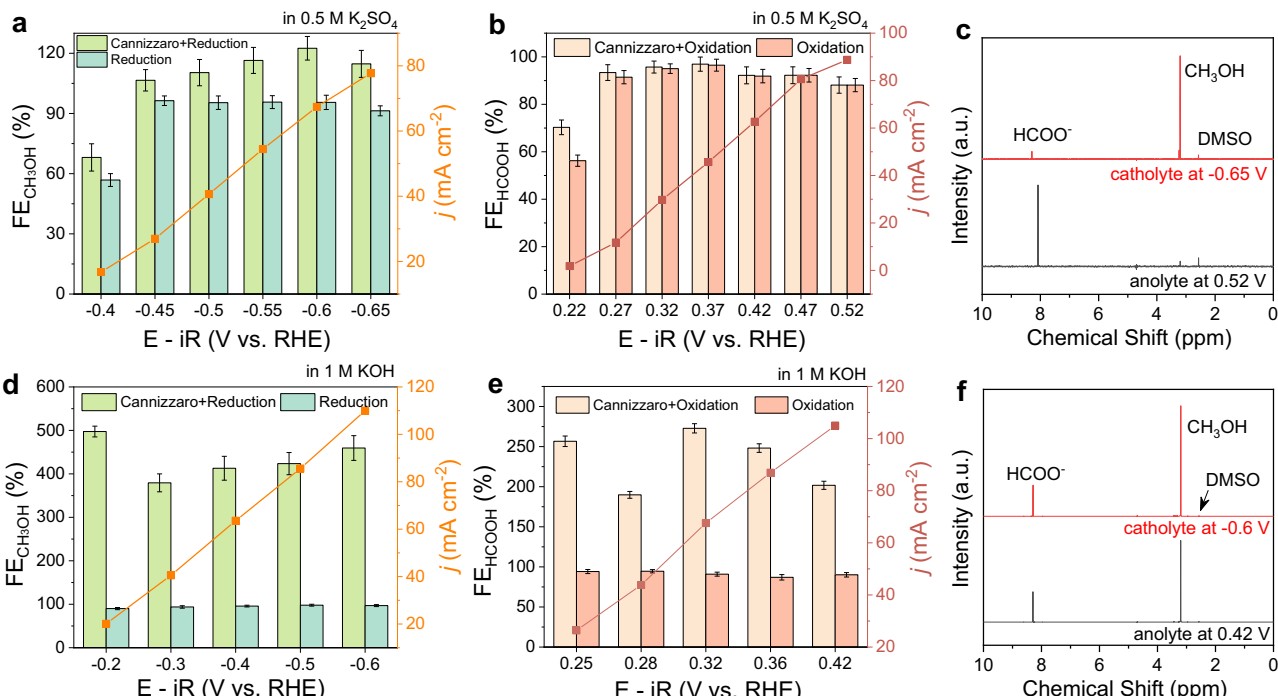

**Fig. 3 | Cannizzaro disproportionation reactions of fresh FA solution in neutral and alkaline electrolytes. a** $FE_{CH3OH}$ and current density of 1 M FA reduction catalyzed by CuTAPc-layer in an H-cell with catholyte: 1 M FA + 0.5 M $K_2SO_4$. **b** $FE_{HCOOH}$ and current density of 1 M FA oxidation catalyzed by $Pt_2Ru$ in an H-cell with anolyte: 1 M FA + 0.5 M $K_2SO_4$. **c** $^1$H NMR spectra of electrolytes from (**a**, **b**). **d** $FE_{CH3OH}$ and current density of 1 M FA reduction catalyzed by CuTAPc-layer in an H-cell with catholyte: 1 M FA + 1 M KOH. **e** $FE_{HCOOH}$ and current density of 1 M FA oxidation catalyzed by $Pt_2Ru$ in an H-cell with anolyte: 1 M FA + 1 M KOH. **f** $^1$H NMR spectra of electrolytes from (**d**, **e**). Each test uses 8 mL fresh electrolyte. The error bars represent the standard deviation of three independent measurements. Source data for the figure are provided as a Source data file.

that generated from electrochemical reduction (Fig. 3d). Disproportionation reactions also take place alongside oxidation processes in alkaline electrolytes (Fig. 3e). Electrolysis experiments were also conducted in flow cells to further evaluate disproportional processes (Figs. S39–S42). Similarly, disproportionation dominates in alkaline environments, which produces 700% higher products than electrochemical conversion. These comparisons highlight the necessity of conducting FA electrolysis in acidic electrolytes for selective transformation.

## Techno-economic analysis of POM waste conversion in acidic electrolytes

The CuTAPc-layer and $Pt_2Ru$ catalysts exhibit high acidic catalytic activity and selectivity for FA reduction and oxidation, respectively, enabling efficient production of methanol and formic acid. Utilizing these catalysts, a two-electrode electrolyzer with a larger active area of 4 cm$^2$ (Fig. 4a) was developed to achieve high single-pass conversion efficiency (Fig. S43). At a cell voltage of 3.5 V, FEs of 91.9 and 90.1% are achieved for methanol and formic acid production, respectively, corresponding to SPCEs of 71.1% and 74.7% towards methanol and formic acid. By further increasing the cell voltage to 4 V, the system delivers a total current of 374.2 mA, and the SPCEs rise to 86 and 89.5% for methanol and formic acid, respectively (Fig. 4b). Figure 4d shows the $^1$H NMR spectra of catholyte and anolyte collected at a cell voltage of 4 V. Note that some concentrated products cross over through the membrane, which could be addressed by membrane or cell optimizations. The concentrations of total methanol and total formic acid are determined to be 0.86 and 0.89 M, respectively. We also examine the SPCE in H-cell configuration for batch-to-batch electrolysis (Figs. S44 and S45). Both catalysts remain efficient in FA conversion, attaining >80% FE and conversion efficiency at both electrodes.

To further emphasize the advantages of acidic electrolysis for POM treatment, techno-economic analysis (TEA; see Figs. S54–S57 and Table S1 for more details) was conducted for three degradation routes using a model adapted from literature[73–76]. For alkaline electrolysis, POM is initially subjected to acidic hydrolysis to obtain concentrated FA solutions, which are then alkalized for electrolysis. Assuming no disproportionation under alkaline conditions, our calculations indicate that electrolysis in alkaline media yields a net profit of $78.2 per ton of POM. This moderate profit is constrained by the high cost of KOH, and is consistent with recent work showing that KOH material cost accounts for more than 40% of the total cost in alkaline systems[77]. However, when disproportionation is taken into account, alkaline electrolysis instead results in a loss of $105/ton POM. Electro-depolymerization in organic solvents, as proposed in a previous study[47], is also employed for the TEA study. In this route, the oxidation of hexafluoroisopropanol (HFIP) to hexafluoroacetone (HFA) generates the protons required for POM depolymerization. Assuming no net loss of HFIP, this approach yields a loss of $3389/ton POM, driven by the high costs of HFIP (required due to POM's poor solubility) and the $LiClO_4$ electrolyte, even with full daily recycling. When the instability and decomposition of oxydimethanol are accounted for, the loss increases to $7342.7/ton POM. In contrast, our strategy for electrocatalytic upgrading of POM in acidic media proves to be economically advantageous, achieving a profit of $215.5/ton POM.

## Mechanistic investigation

To delve into the mechanistic origin of high activity in acids, we use in situ attenuated total reflectance surface-enhanced infrared absorption spectroscopy (ATR-SEIRAS) to probe the surface species (Fig. 5a). For FA reduction, in situ ATR-SEIRAS spectra of CuTAPc and CuTAPc-layer are presented in Figs. 5a and S46. FA undergoes addition reaction with water molecules to form methanediol[78], which shows C-H

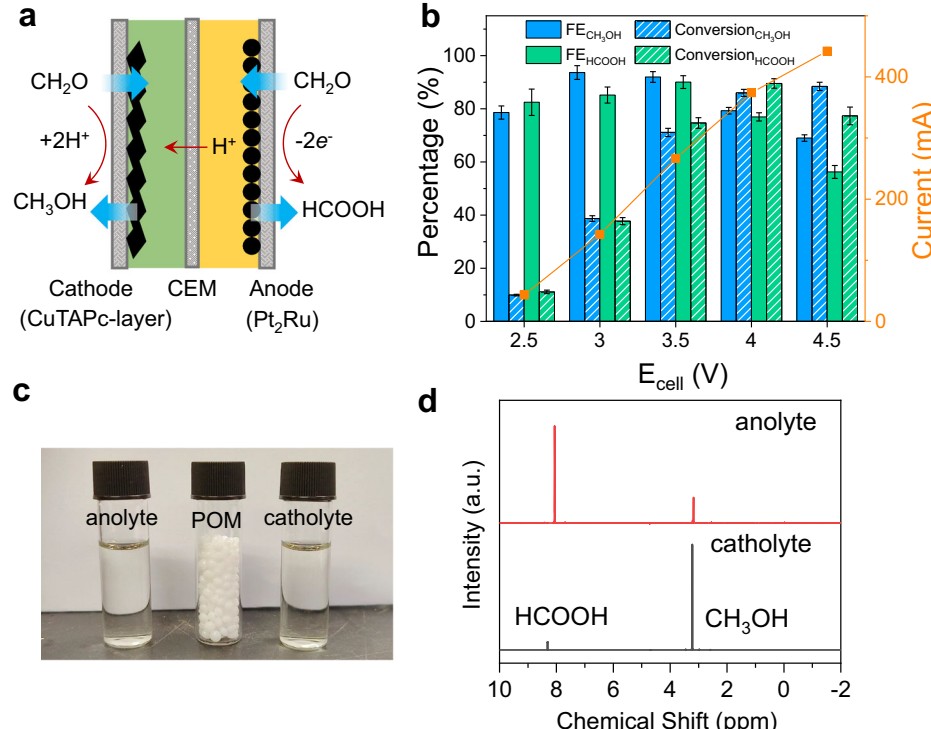

**Fig. 4 | Electrocatalytic synthesis and economic evaluation of POM waste conversion in acidic electrolytes. a** Schematic of FA reduction and oxidation in acidic media in a flow cell. **b** Single pass conversion efficiencies, Faradaic efficiencies, and currents at different cell voltages, catholyte: 1 M FA + 0.5 M $K_2SO_4$ + 0.05 M $H_2SO_4$, anolyte: 1 M FA + 0.5 M $K_2SO_4$ + 0.5 M $H_2SO_4$. The electrocatalytic performance was measured without iR compensation. The error bars represent the standard deviation of three independent measurements. **c** Photos of POM, along with catholyte and anolyte, taken after electrolysis. **d** $^1$H NMR spectra of anolyte and catholyte at a cell voltage of 4 V from (**b**). Source data for the figure are provided as a Source data file.

bending vibrations and C-O stretching vibrations at 1433, 1370, 1284, and 1120 cm$^{-1}$. A new band at 1032 cm$^{-1}$ emerges after applying negative potentials, corresponding to the C-O vibration of the CH$_3$O* intermediate[79,80].

The hydrogen bonding network at the electrical double-layer is crucial for HER activity. Interfacial water with a lower degree of hydrogen bonding decreases the energy barrier for water dissociation, promoting the HER process[81,82]. The interfacial water structures of CuTAPc and CuTAPc-layer are analyzed in Figs. 5b, c and S47. Adsorption peaks at 3250, 3450, and 3600 cm$^{-1}$ are ascribed to 4-coordinated hydrogen-bonded water (4-HB-H$_2$O), 2-coordinated hydrogen-bonded water (2-HB-H$_2$O), and cation-hydrated water with weak hydrogen bonding interactions (K$^+$-H$_2$O), respectively. For the CuTAPc-layer, the content of 4-HB-H$_2$O remains relatively constant at nearly 40% (Fig. 5c). In contrast, the content of 4-HB-H$_2$O over the original CuTAPc is continuously consumed and decreases to about 20% at −1.4 V (Fig. 5b). Additionally, CuTAPc shows a steeper Stark slope of K$^+$-H$_2$O and a higher proportion of K$^+$-H$_2$O, suggesting a stronger interaction between K$^+$-H$_2$O and CuTAPc (Fig. S48). This stronger interaction supports the enhanced HER activity observed for CuTAPc. Rotating disk electrode experiments[83,84] were carried out in Ar-saturated 0.5 M K$_2$SO$_4$ + 0.05 M H$_2$SO$_4$ to further compare the hydrogen evolution activity, with detailed current curves shown in Fig. S49. The CuTAPc-layer catalyst exhibits a lower current density than CuTAPc over the range of −0.5 to −1.25 V. A current-density plateau is observed, which we attribute to the depletion of hydronium ions. The CuTAPc-layer presents a smaller plateau current density. Linear fitting of the currents at −1.23 V using the Levich equation indicates a lower H$_3$O$^+$ diffusion coefficient, D$_{H3O+}$, for the CuTAPc-layer. Furthermore, kinetic current densities calculated via the Koutecký-Levich equations show that the CuTAPc-layer demonstrates a decreased HER current density of 203.5 mA cm$^{-2}$, compared to 301.2 mA cm$^{-2}$ for CuTAPc.

These results corroborate that the hydrophobic alkyl group grafted cationic CuTAPc-layer displays suppressed HER activity, which competes with FA reduction in acidic solution.

The above analyses suggest that the reduction mechanism, depicted in Fig. 5d, follows the pathway of adsorption, with subsequent formation of CH$_2$O*, CH$_3$O*, and finally CH$_3$OH[85,86]. However, for CuTAPc, only the signal of methanediol is observed (Fig. S46), possibly due to the high HER activity. This interpretation is supported by the interfacial water structure, where the dominant weakly bonded water for CuTAPc favors the HER side reaction, and the dominant strongly bonded water for CuTAPc-layer suppresses HER.

For FA oxidation, we first use in situ ATR-SEIRAS to probe the evolution of surface species on Pt$_2$Ru at varying potentials (Fig. 6a). The data reveal adsorption bands at 1434 and 1282 cm$^{-1}$, which are attributed to the C-H and C-O vibrations of HOCH$_2$O* intermediate. At higher overpotentials, peaks at 1703, 1563, and 1364 cm$^{-1}$ become more intense, which are indexed to the formic acid intermediate[87–89].

To further understand the different activities observed across different metal catalysts, we used the free energies calculated by density functional theory (DFT) at 0.6 V, pH 1 (see "Methods" for details). Figure 6b shows the structure of solvated FA on different metal surfaces. The metal-O bond is summarized in Fig. 6c, which provides information on the strength of metal-FA interaction and a qualitative description for FA activation. The bond distances decrease in the order of Ag (2.88 Å) > Au (2.77 Å) > Pt (2.57 Å) > Pd (2.16 Å) > Ru (2.09 Å) > Pt$_2$Ru (2.05 Å). The shorter bond distances on Ru-containing surfaces, particularly Pt$_2$Ru, appear consistent with the earlier onset potential and higher partial currents for HCOOH production, suggesting that stronger FA activation may be facilitated. In contrast, Au, Ag, and Pt likely bind too weakly, making the initial FA activation unfavorable.

To deepen our understanding, the free-energy profiles for the full reaction steps on different metal slabs were computed (Fig. 6d). We

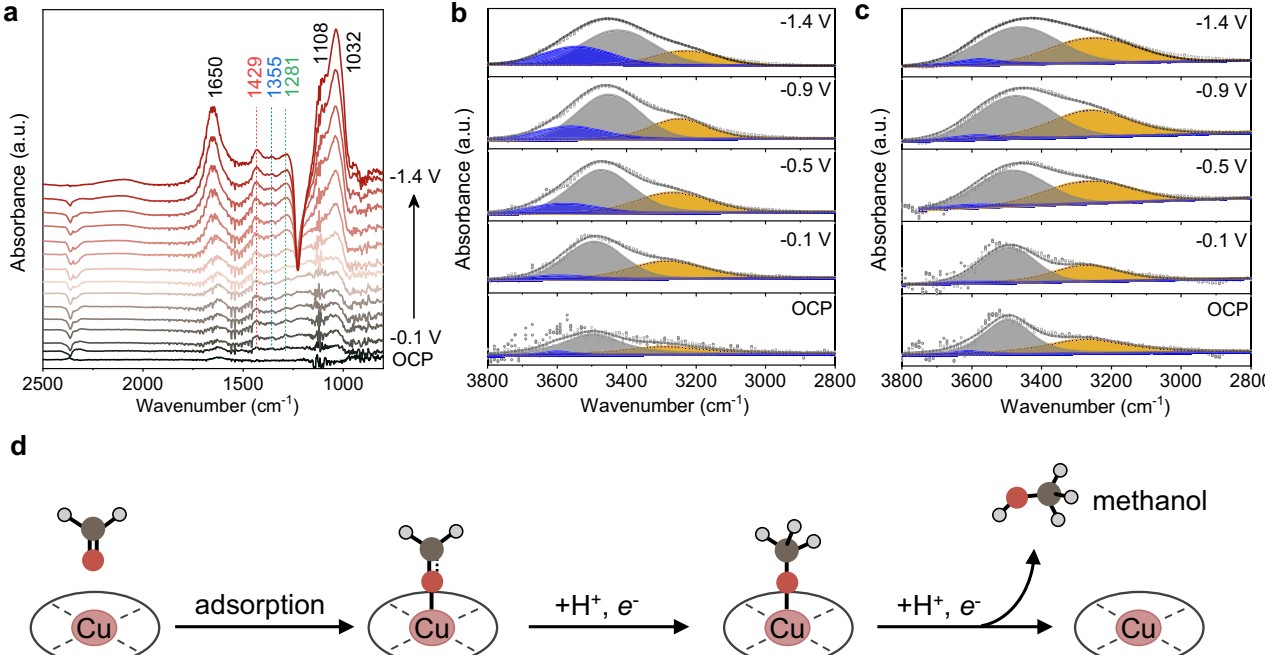

**Fig. 5 | Mechanistic investigations of acidic FA reduction. a** In situ ATR-SEIRAS spectra of electrocatalytic reduction process catalyzed by CuTAPc-layer in 1 M FA + 0.5 M $K_2SO_4$ + 0.05 M $H_2SO_4$, OCP: Open Circuit Potential. Deconvolution of interfacial water structure during in situ ATR-SEIRAS of electrocatalytic reduction process for **b** CuTAPc and **c** CuTAPc-layer. The yellow, gray, and blue shaded areas correspond to 4-HB-$H_2O$, 2-HB-$H_2O$, and $K^+$-$H_2O$, respectively. **d** Schematic diagram of FA reduction mechanism. The dark gray, red, and gray spheres represent carbon, oxygen, and hydrogen atoms, respectively. Source data for the figure are provided as a Source data file.

see that $Pt_2Ru$ and Ru pull every intermediate far more downhill than for Ag, Au, Pt, and Pd. Initial activation and adsorption is exergonic on $Pt_2Ru$, Ru, and Pd, but only weakly so on Ag, Pt, and Au. Formation of *$OCH_2OH$ from *$OHCH_2OH$ further exacerbates the FA conversion, shrinking the effective thermodynamic span that a kinetic barrier bridges. The *$OHCH_2OH$ deprotonation is uphill by approximately 1 V for Ag, Au, and Pt, whereas the barrier reduces to around 0.5 V for $Pt_2Ru$ and Ru. After this step, the second deprotonation yields *OCHOH, which is downhill for all metals.

The above in situ spectroscopy and computation suggest that FA oxidation follows a sequential mechanism (Fig. 6e): in the first step, methanediol is adsorbed to form *$OCH_2OH$ with the loss of proton, which is further oxidized to *OCHOH and ultimately desorbs as HCOOH. In regard to the physical reasons as to why $Pt_2Ru$ and Ru performs better than other metals, the oxophilicity and interfacial kinetics of the metal surfaces lead to enhanced stability at the catalytic sites. Ru is more oxophilic than Pt/Au/Ag, and forms surface -OH at lower potentials. In acid, formation of HCOOH occurs through a gem-diol route that requires proximal OH. The Ru sites decrease the free-energy and reorganization penalties for the favorable *$OHCH_2OH$ and *$OCH_2OH$ formation as compared to other catalysts. In the $Pt_2Ru$ alloy, adjacent dissimilar atoms create cooperative sites, where Ru provides oxophilic sites for stabilization, and Pt offers less oxophilic sites that favor rearrangement and release. This reduces the thermodynamic burden on the surface for absorption and desorption, with the mix of oxophilic and non-oxophilic metals decreasing both the dehydrogenation barrier and thermodynamic span in the rate-determining step of release following absorption onto the surface. It is recognized that the DFT calculations were performed on simplified model systems under idealized conditions, which may not fully capture the complex electrochemical environment. Yet, the computational findings offer a valuable theoretical framework that aligns with and supports the experimental observations.

## Discussion

In summary, we report a selective and efficient strategy for the electrochemical upcycling of POM waste into valuable methanol and formic acid products in acidic electrolytes. The CuTAPc-layer/$Pt_2Ru$ system demonstrates high electrocatalytic activity, achieving Faradaic efficiencies of 93.2 and 91.3% for methanol and formic acid at the cathode and anode, respectively. The coupled electrolyzer also achieves favorable single-pass conversion efficiencies of 86% and 89.5% for FA-to-$CH_3OH$ and FA-to-HCOOH conversions, with a total current of 374.2 mA at a cell voltage of 4 V. The acidic system mitigates the Cannizzaro disproportionation, which results in significant FA loss, diminished selectivity, and complex downstream purification. Thus, effectively suppressing these parasitic pathways through tailored reaction environments will enable higher FA conversion efficiency and improved product purity. Moreover, our strategy operates at ambient conditions, which addresses the formaldehyde leakage issues commonly associated with conventional thermal disposal methods. Our acidic FA reduction and oxidation can also be coupled with other electrochemical reactions for paired electrolysis, such as $CO_2$ reduction and biomass oxidation. Overall, this work establishes an eco-friendly pathway for POM waste upcycling, offering an alternative for sustainable waste management and chemical production. This motivates the development of new catalysts to further enhance acidic POM waste upcycling. Our electrochemical disproportionation method can be extended to other substrates with suitable redox potentials, such as hypochlorite disproportionation for toxic chemical mediation.

## Methods
### Materials
All chemicals were purchased and used without further purification. Ammonium acetate (98%), N,N-dimethylacetamide (DMAc, 99.5%), paraformaldehyde (97%), acetic acid (Hac, 99.8%), N-methyl-2-pyrrolidone (NMP, 99.5%), formaldehyde solution (37 wt% in $H_2O$), and methyl iodide (99%) were obtained from J&K Chemical Ltd. 2,5-

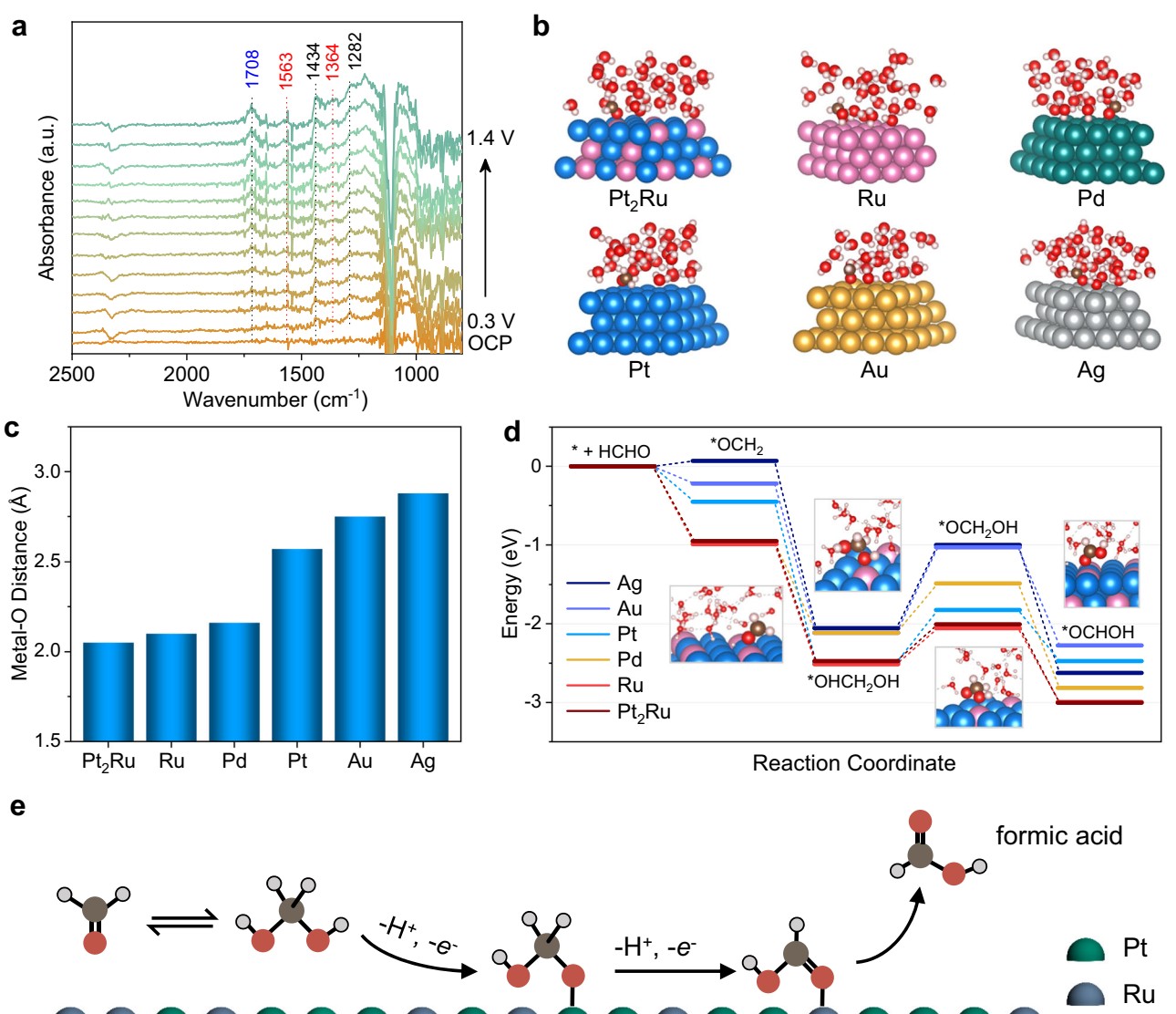

**Fig. 6 | Mechanistic investigations of acidic FA oxidation. a** In situ ATR-SEIRAS spectra of electrocatalytic oxidation process catalyzed by Pt₂Ru in 1 M FA + 0.5 M K₂SO₄ + 0.5 M H₂SO₄. **b** Adsorption geometry of FA on different metal surfaces and **c** A summary of metal-OCH₂ distances for different metals. The blue, pink, green, yellow, and gray spheres represent platinum, ruthenium, palladium, gold, and silver atoms, respectively. **d** Calculated energy pathway for Ag, Au, Pd, Pt, Ru, and Pt₂Ru at U = 0.6 V, pH = 1. Inserts show the corresponding intermediates adsorbed on Pt₂Ru. **e** Schematic diagram of FA oxidation mechanism. The dark gray, red, gray, cyan, and blue spheres represent carbon, oxygen, hydrogen, platinum, and ruthenium atoms, respectively. Source data for the figure are provided as a Source data file.

ditert-butyl-1,4-benzoquinone (98%) and 1-iodohexane (98%) were provided by Tokyo Chemical Industry Co., Ltd. (TCI). Ethanol (99.7%), isopropyl alcohol (99.5%), ethyl acetate (99.5%), methanol (99.5%), ethyl ether (99%), hydrochloric acid (37%), sulfuric acid (99.8%), dimethyl sulfoxide (DMSO, 99.5%), and N,N-dimethylformamide (DMF, 99.5%) were supplied by AQA, USA. Formic acid (99%), potassium hydroxide (95%), potassium sulfate (99%), potassium chloride (99.5%), and acetyl acetone (99.5%) were obtained from Aladdin (Shanghai). Ruthenium (III) chloride trihydrate (98%), hydrogen hexa-chloroplatinate (99.9%), sodium borohydride (99%), and 5 wt% Nafion solution were purchased from Sigma-Aldrich. 20% Pd/C and 40% Au/C were purchased from SCI Materials Hub. Polyoxymethylene, Cu (99.9%), Cu₂O (99.9%), and Ag powder (99.9%) were obtained from Shanghai Macklin Biochemical Co., Ltd. Deuteroxide (Cambridge Isotope Laboratories, 99.9%), copper tetraamionphthalocyanine (Shanghai Kaiyulin Pharmatech Co., Ltd, 95%), anion exchange membrane (FAB-PK-130, FuMA-Tech), cation exchange membrane (Nafion N117,

Fuel Cell Store), carbon black (Cabot, XC 72R), and gas diffusion layer (YLS 30T, Suzhou Sinero Technology) were obtained from the corresponding reagent company. Deionized water (resistivity: 18.2 MΩ/cm) was used for all the experiments. The pretreatment for Nafion N117 membrane involves sequential heating in 5% H₂O₂ and 1 M H₂SO₄ solutions at 80 °C to remove organic/inorganic impurities and convert it to the H⁺ form, followed by thorough rinsing with deionized water.

## Preparation of catalysts

**Synthesis of CuTAPc-layer.** 4 mg CuTAPc, 20 mg 2,5-ditert-butyl-1,4-benzoquinone, 5 mL dimethylacetamide, 2 mL ethanol, and 0.2 mL of 6 M acetic acid solution were degassed in a 20 mL Pyrex tube by three freeze-pump-thaw cycles. The tube was sealed and heated at 120 °C for 3 days. The resulting imine precipitate was collected by centrifugation, washed with dimethylformamide and ethanol to remove residues, and dried under vacuum at 60 °C overnight. Next, the mixture of 30 mg obtained imine solid, 0.02 mL 1-iodohexane, and 10 mL of

N-methylpyrrolidone was degassed in a 50 mL Pyrex tube by three freeze-pump-thaw cycles. The tube was sealed and heated at 120 °C for 3 days. After the reaction, the product was washed with ethyl acetate and vacuum-dried at 60 °C. The obtained solid was further treated with methyl iodide to prepare fully quaternized catalysts. Subsequently, 30 mg of precipitate was treated with 0.2 mL methyl iodide in a 25 mL round bottom flask and heated at 50 °C for 12 h. After the reaction, the final product was precipitated by adding diethyl ether into the mixture and washed with ethanol. Finally, the product was freeze-dried to yield loose powder.

**Synthesis of Pt$_2$Ru alloy.** Ruthenium (III) chloride trihydrate (6.4 mg; 0.0245 mmol) and hydrogen hexachloroplatinate (IV) (20 mg; 0.0488 mmol) were dissolved in 10 mL deionized water. After stirring for 30 min, 3 mL of 1 mg/mL NaBH$_4$ solution was added. The mixture was stirred at room temperature (25±2 °C) for 1 h. The resulting product was collected by centrifugation, washed with isopropyl alcohol and deionized water, and then freeze-dried. For alloys with different compositions, the precursor ratios were adjusted according to the desired stoichiometry while maintaining the total metal content at 0.73 mmol.

## Characterization

X-ray diffraction analysis was carried out using Bruker D2. Transmission electron microscopy imaging of the catalysts was performed on Philips Technai 12 and JEM-2100F equipped with EDS detector. XPS spectra were recorded on Thermo Scientific K-Alpha equipped with an Al X-ray excitation source (1486.6 eV). All binding energies were referenced to the C1s peak at 284.8 eV. Solid state $^{13}$C nuclear magnetic resonance spectra were test by Bruker Avance III 600. Atomic force microscopy measurement was performed using Bruker Icon in tapping mode. Fourier transform infrared spectroscopy was recorded using KBr pellets on PerkinElmer Spectrum 100 spectrometer in the range of 500–4000 cm$^{-1}$. UV-vis spectra were collected on Shimadzu-UV1700. Contact angle measurements were conducted using Sindin SDC-350KS. Zeta potentials were measured on a dynamic light scattering particle size analyzer (Malvern Zetasizer Nano-ZS). ICP-OES analysis was conducted on PerkinElmer AVIO 220 MAX spectrometer.

## Electrochemical measurements

**FA solution.** 15 g paraformaldehyde was added to 250 mL 0.5 M H$_2$SO$_4$ and heated at 40 °C for 3 h until fully dissolved. For POM depolymerization, the 6.8 g POM pellets were ground into powder, which completely dissolved in 200 mL of 0.5 M H$_2$SO$_4$ at 90 °C within 2 h. The concentrated FA solution was obtained by digesting 75 g of grounded POM powder in 150 mL of 0.5 M H$_2$SO$_4$ at 90 °C overnight. The decomposed FA solutions were stored at 4 °C.

**H-cell.** Electrochemical measurements were performed in an 8 mL glassy H-type cell. The cathode and anode chambers were separated by a cation exchange membrane (Nafion N117). The mass loading of the CuTAPc-layer/Pt$_2$Ru catalyst on gas diffusion layer (GDL) is 1 mg cm$^{-2}$. To prepare cathode: 2 mg CuTAPc-layer, 2 mg carbon black, and 20 μL Nafion solution were dispersed in 1 mL ethanol and sonicated for 15 min. 0.5 mL CuTAPc-layer ink was drop-casted onto a GDL with an area of 1 cm$^2$. For the anode: 2 mg Pt$_2$Ru and 20 μL nafion solution were dispersed in 1 mL ethanol and sonicated for 15 min. 0.5 mL Pt$_2$Ru ink was drop-casted onto a GDL with an area of 1 cm$^2$. The counter and reference electrodes used were a platinum foil and a Hg/Hg$_2$SO$_4$ electrode, respectively. Chronoamperometric curves were recorded on Ivium electrochemical working station. For FA reduction: the catholyte was composed of 1 M FA + 0.5 M K$_2$SO$_4$ + 0.05 M H$_2$SO$_4$, and anolyte was 0.5 M H$_2$SO$_4$. The catholyte was prepared by diluting predecomposed, concentrated FA stock solution to the target concentration with the specified electrolyte salts. For FA oxidation:

anolyte was composed of 1 M FA + 0.5 M K$_2$SO$_4$ + 0.5 M H$_2$SO$_4$, and catholyte was 0.5 M H$_2$SO$_4$. The anolyte was achieved by directly dissolving the electrolyte salts in a low-concentration FA aqueous solution. The electrolyte solutions were freshly prepared every day. All potentials in the H-cell experiments were iR-corrected (85%) and converted to the RHE scale via $E_{RHE} = E_{Hg/Hg_2SO_4} + 0.0591 \times pH + 0.656 \text{ V} - 85\% \times i \times R$. The calibration of Hg/Hg$_2$SO$_4$ reference electrode was operated using a standard three-electrode system with polished platinum foil as the working and counter electrodes in an electrolyte (0.5 M H$_2$SO$_4$) continuously purged with hydrogen. Cyclic voltammetry (CV) was subsequently performed at a slow scan rate of 1 mV s$^{-1}$. The mean value of the potentials where the current crossed zero during the CV cycles was defined as thermo-dynamic potential for the hydrogen electrode reactions. In our experiment, in 0.5 M H$_2$SO$_4$, the zero current point was calculated to be 0.656 V. The electrolytes were characterized by $^1$H NMR (Bruker AVANCE AV III 300) with dimethyl sulfoxide as the internal standard. For the $^1$H NMR analysis, 450 μL of the electrolyte was mixed with 50 μL of a 10 mM dimethyl sulfoxide solution in D$_2$O.

**Flow cell.** 50 mL catholyte and anolyte were circulated at a flow rate of 3 mL min$^{-1}$ by a peristaltic pump, respectively. The cathode and anode chambers were separated by a cation exchange membrane (Nafion N117). Both the cathode and anode were prepared using the same method as in the H-cell, with a catalyst loading of 1 mg cm$^{-2}$, except using a GDL with an area of $1 \times 2.5$ cm$^2$. The working area was $0.5 \times 2$ cm$^2$. The composition of catholyte was 1 M FA + 0.5 M K$_2$SO$_4$ + 0.05 M H$_2$SO$_4$, and anolyte was 1 M FA + 0.5 M K$_2$SO$_4$ + 0.5 M H$_2$SO$_4$. For the expanded area electrolyzer, the catalysts were coated on a $2.5 \times 2.5$ cm$^2$ GDL to achieve a catalyst loading of 1 mg cm$^{-2}$. The electrocatalytic performance in the integrated full cell was measured without iR compensation.

In the H-cell tests, an 85% iR compensation was applied to mitigate the solution resistance effect. The detailed uncompensated resistance values for each catalyst are listed in Supplementary Note 5. In contrast, all performance evaluations for the full flow cell were conducted without iR compensation to reflect practical operating conditions.

**Faradaic efficiency.** The Faradaic efficiencies for FA conversion were calculated according to:

$$FE_{CH3OH} = \frac{2 * 96485 \, C/mol * n_{CH3OH}}{total \, charge} * 100\% \qquad (3)$$

$$FE_{HCOOH} = \frac{2 * 96485 \, C/mol * n_{HCOOH}}{total \, charge} * 100\% \qquad (4)$$

$$FE_{H_2} = \frac{2 * 96485 \, C/mol * n_{H_2}}{total \, charge} * 100\%$$
$$= \frac{2 * 96485 \frac{C}{mol} * \frac{P}{RT}}{total \, charge} * v * t * 100\% \qquad (5)$$

P is the atmospheric pressure, T is the temperature, R is the molar gas constant, and $v$ is the gas flow rate.

**Single pass conversion efficiency.** The SPCEs for FA conversion were calculated according to:

$$SPCE_{CH3OH} = \frac{n_{CH3OH}}{n_{FA}} * 100\% \qquad (6)$$

$$SPCE_{HCOOH} = \frac{n_{HCOOH}}{n_{FA}} * 100\% \qquad (7)$$

**Disproportionation reaction.** In neutral media, for FA reduction, the catholyte was composed of 1 M FA + 0.5 M $K_2SO_4$, and the anolyte was 0.5 M $K_2SO_4$. For FA oxidation, the anolyte was composed of 1 M FA + 0.5 M $K_2SO_4$, and the catholyte was 0.5 M $K_2SO_4$. In alkaline media, for FA reduction, the catholyte was composed of 1 M FA + 1 M KOH, and the anolyte was 1 M KOH. For FA oxidation, the anolyte was composed of 1 M FA + 1 M KOH, and the catholyte was 1 M KOH. The Faradaic efficiencies were calculated according to:

$$FE_{Cannizzaro+Reduction} = \frac{2 * 96485\,C/mol * n_{CH3OH,total}}{total\ charge} * 100\% \quad (8)$$

$$FE_{Reduction} = \frac{2 * 96485\,C/mol * (n_{CH3OH,total} - n_{HCOOH})}{total\ charge} * 100\% \quad (9)$$

$$FE_{Cannizzaro+Oxidation} = \frac{2 * 96485\,C/mol * n_{HCOOH,total}}{total\ charge} * 100\% \quad (10)$$

$$FE_{Oxidation} = \frac{2 * 96485\,C/mol * (n_{HCOOH,total} - n_{CH3OH})}{total\ charge} * 100\% \quad (11)$$

**RDE experiments.** The RDE electrode was prepared as follows: 2 mg CuTAPc-layer, 2 mg carbon black, and 20 μL of 5 wt% Nafion solution were combined in 1 mL ethanol. This mixture was sonicated for 15 min. Then, 10 μL ink was drop-cast onto the RDE (Pine Research) with a diameter of 5 mm and dried at room temperature (25±2 °C). The electrolyte used was 0.5 M $K_2SO_4$ + 0.05 M $H_2SO_4$ and saturated with argon. LSV curves were recorded at a scan rate of 10 mV/s.

**In situ ATR-SEIRAS measurements.** In situ ATR-SEIRAS spectroscopy measurements were conducted using a Nicolet iS50 FTIR spectrometer equipped with an HgCdTe (MCT) detector cooled with liquid nitrogen. The Au-coated Si semi-cylindrical prism (60°, 20 × 0.95 mm) was used as the conductive substrate for catalysts and the IR reflection element. The catalyst inks were prepared by mixing 5 mg of electrocatalyst, 5 mg of carbon black, 50 μL of Nafion solution, and 10 mL of ethanol. 1 mL of this ink dispersion was then carefully dropped onto an Au film-coated Si prism and dried in air. The ATR-SEIRAS measurements were performed by recording 32 scans at a spectral resolution of 4 cm$^{-1}$. The spectrum collected under open circuit voltage was used as the background.

**Computation methods**
Computational tools involved in modeling these systems involved using spin-polarized periodic density functional theory (DFT) as implemented in the Vienna ab initio simulation package (VASP 6.4.2) for geometry optimization and phonon calculations. We used the Perdew-Burke-Emzerhof (PBE) functional with a plane-wave cutoff energy of 500 eV[90–95]. We used the D3(BJ) empirical van der Waals interactions. For structural optimizations, the Brillouin zone was sampled using a 2 × 2 × 1 K-point grid based on the Gamma scheme. The convergence criteria for forces were set to 0.02 eV/A, while electronic structure energy convergence was $10^{-6}$ eV.

To accurately represent the aqueous environment of the electrochemical cell, the model incorporates the influence of three water layers (24 water molecules) on the thermodynamic stability of adsorbed intermediate species. Solvation can lower the activation barrier for proton transfer steps, especially when formate interacts with water clusters. A trilayer of water mimics the interfacial structure under anodic conditions, where strong hydration and electric double-layer effects dominate[96]. The optimized structures can be found in Supplementary Data. Hydrogen bonding interactions with water significantly enhance the stability of surface-bound intermediates.

All DFT calculations were performed at zero applied potential (U = 0 V) and at 0 pH within the framework of the computational hydrogen electrode (CHE) model. To account for the effect of applied electrode potential on the free energy of electron transfer steps, a voltage correction was introduced. The electrochemical potential of an electron at potential U is given by: $\mu_e = -eU$, where e is the elementary charge, and U is the electrode potential referenced to the CHE. Consequently, for any elementary step involving $n_e$ electrons, the corrected Gibbs free energy is expressed as:

$$\Delta G(U) = \Delta G_{DFT} - n_e eU \quad (12)$$

Here, $\Delta G_{DFT}$ includes zero-point energy ($\Delta ZPE$) and entropic corrections ($T\Delta S$) evaluated at U = 0 V. This approach effectively shifts the free energy of proton–electron transfer steps to reflect the applied potential, enabling accurate construction of potential-dependent free energy diagrams. Positive potentials stabilize oxidation steps, while negative potentials favor reduction steps. All reported energetics at non-zero potentials were obtained using this correction.

To incorporate pH effects, the free energy of a proton is adjusted by:

$$\Delta G_{pH} = k_B T\ln(10) \times pH \approx 0.059 \times pH\ (in\ eV\ at\ 298\ K) \quad (13)$$

Thus, the total free energy correction for a reaction step involving $n_e$ electrons is:

$$\Delta G(U, pH) = \Delta G_{DFT} + \Delta ZPE - T\Delta S - n_e eU + n_H \times 0.059 \times pH \quad (14)$$

This ensures that both applied potential and electrolyte acidity are properly reflected in the computed energetics[97]. The oxidation of formaldehyde to formate is thermodynamically favored because positive potential stabilizes electron-removal steps. However, at pH 1, the high proton activity minimizes the destabilizing effect on proton-coupled electron transfer, ensuring that the formation of HCOOH remains energetically accessible under acidic conditions.

## Data availability
Source data are provided with this paper.

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

## Acknowledgements

R.Y. acknowledges support from the National Natural Science Founda-tion of China (22522509), Guangdong Basic and Applied Basic Research Fund (2024A1515030164), Hong Kong Research Grant Council

(11309723, 11310624), the Shenzhen Science and Technology Program (JCYJ20220818101204009), and State Key Laboratory of Marine Environmental Health (SKLMP/SCRF/0060). B.Z.T. acknowledges support from National Key Research and Development Program of China (2023YFB3810001), National Natural Science Foundation of China (52333007), Key-Area Research and Development Program of Guangdong Province (2024B0101040001), Shenzhen Key Laboratory of Functional Aggregate Materials (ZDSYS20211021111400001), the Science Technology Innovation Commission of Shenzhen Municipality (KQTD20210811090142053 and JCYJ20220818103007014), and Innovation and Technology Commission (ITC-CNERC14SC01). W.A.G. thanks the Liquid Sunlight Alliance, which is supported by the U.S. Department of Energy, Office of Science, Office of Basic Energy Sciences, Fuels from Sunlight Hub under Award Number DE-SC0021266, for supporting the Caltech expenses.

## Author contributions

R.Y. proposed and designed the experiment. R.Y., B.T., W.A.G., and M.R. supervised the project. Y.S. performed the catalyst preparation and the catalytic tests. Z. Z. helped with the characterizations. J.S., W.G., Y.L., G.L., Y.X., Q.Z., M.H., R.W., and R.X. contributed to data analysis. X.W. helped with FTIR characterizations. C.X. and S.Z. discussed the results and commented on the manuscript. T.D., A.D.R., and W.A.G. carried out the DFT calculations and analysis. All authors contributed to the discussion and manuscript preparation.

## Competing interests

The authors declare no competing interests.
