## [Transparent Peer Review file · Nature Communications]

Electrochemically mediated disproportionation for selective formaldehyde upcycling in acid

Corresponding Author: Professor Ruquan Ye

Version 0:

Reviewer comments:

Reviewer #1

(Remarks to the Author)

The authors have made revisions in response to the reviewers' comments; however, the central novelty of the work remains unconvincing.

1. Electrooxidation and electroreduction of formic acid (FA) in acidic media have been extensively investigated for decades (e.g., DOI: 10.1016/0022-0728(82)85084-5; 10.1016/0022-0728(91)85259-R; 10.1021/la0200459; 10.1016/j.jelechem.2003.09.014; 10.1016/0368-1874(85)85762-2; 10.1016/j.jcis.2009.01.020). The present study appears to mainly integrate FA oxidation and reduction processes using well-established catalytic systems, without introducing a fundamentally new reaction pathway, catalyst concept, or mechanistic insight.
2. The authors claim that conventional alkaline FA oxidation suffers from disproportionation-induced FA loss and complicated downstream separation, yet the proposed system still requires additional separation steps and generates by-products. It is therefore unclear how the new process offers a substantive advantage in terms of product purity, process simplicity, or overall efficiency.
3. The manuscript reports DFT calculations conducted at 0.6 V and pH 1, but the computational methodology provided does not adequately describe how electrode potential and proton activity were explicitly incorporated. Without a clear constant-potential framework or justified approximations, the reliability and physical relevance of these calculations remain questionable.
4. In TEA, prior studies have demonstrated that noble-metal-free catalysts can achieve competitive FA conversion in alkaline electrolytes, whereas the present work employs precious-metal-based anode catalysts, which substantially increases material costs. The manuscript does not sufficiently justify this choice, nor does it explain how catalyst cost and scalability were rationally accounted for in the TEA, raising concerns about the economic realism of the analysis.

Reviewer #2

(Remarks to the Author)

Authors have well addressed my concerns. I have no additional questions.

Reviewer #3

(Remarks to the Author)

The author has satisfactorily addressed the issues raised by the reviewers. It is recommended that the manuscript be accepted.

Version 1:

Reviewer comments:

Reviewer #1

(Remarks to the Author)

I have no further comments

Reviewer #1 (Remarks to the Author):

The authors have made revisions in response to the reviewers' comments; however, the central novelty of the work remains unconvincing.

Reply: We thank the reviewer for the recognition of our previous response and for the valuable comments regarding novelty. The following are detailed responses to comments:

1. Electrooxidation and electroreduction of formic acid (FA) in acidic media have been extensively investigated for decades (e.g., DOI: 10.1016/0022-0728(82)85084-5; 10.1016/0022-0728(91)85259-R; 10.1021/la0200459; 10.1016/j.jelechem.2003.09.014; 10.1016/0368-1874(85)85762-2; 10.1016/j.jcis.2009.01.020). The present study appears to mainly integrate FA oxidation and reduction processes using well-established catalytic systems, without introducing a fundamentally new reaction pathway, catalyst concept, or mechanistic insight.

Reply: We thank the reviewer for the valuable comment and for providing the relevant references on formaldehyde (FA) conversion. We agree that electrocatalytic formaldehyde oxidation and reduction have been investigated over the past decade. Our work, however, is not a simple reuse of these systems, but focuses on the upgrading polyoxymethylene waste under acidic conditions, combining: effective upgrading, a nearly complete conversion rate, and the avoidance of alkaline disproportionation.

Previous studies on electrooxidation and electroreduction of formaldehyde in acidic media predominately investigate the mechanism and reaction pathways of formaldehyde conversion. Incorporating the insights from these reported works, we propose the reaction mechanism for our system. **However, upon careful reading of these papers, the catalysts in previous work show only moderate catalytic activity and limited conversion ratios. In addition, some catalysts show no electro-oxidation activity toward formaldehyde in acidic media (10.1016/0368-1874(85)85762-2), which further highlights the challenge and underscores the significance of achieving efficient polyoxymethylene upgrading under acidic conditions in our work.** Thus, the scope of study, the type of catalysts, the performance, and the significance of our work differ substantially from literature.

2. The authors claim that conventional alkaline FA oxidation suffers from disproportionation-induced FA loss and complicated downstream separation, yet the proposed system still requires additional separation steps and generates by-products. It is therefore unclear how the new process offers a substantive advantage in terms of product purity, process simplicity, or overall efficiency.

Reply: We thank the reviewer for the insightful comment. The disproportionation in alkaline media limits the formaldehyde conversion and yields a mixture that is intrinsically harder to separate. As shown in Supplementary Note 1, even at a low formaldehyde concentration (0.1 M FA + 1 M KOH), 35.8% of HCHO is lost to disproportionation within 3 h. The resulting mixture of methanol and formate in KOH-containing electrolytes complicates downstream processing, since isolation of formic acid and methanol requires (i) acidification to convert formate to formic acid, followed by (ii) further separation steps. Given the higher price of KOH to K₂SO₄/KCl, this acidification is step is economically unfavorable. According to our TEA study (Supplementary Note 4, Case B and C), the input of KOH accounts for more than 40% of the total costs, consistent with reported analyses of alkaline systems (Science 372, 1074-1078 (2021); Nat. Catal. 8, 771-783 (2025)).

In contrast, in acidic media, the products are inherently separated into two streams, with methanol + K₂SO₄ in the catholyte, and formic acid + K₂SO₄ in the anolyte. In each compartment, K₂SO₄

serves as a supporting electrolyte that can be readily crystallized and recycled, and the target products (formic acid and methanol) can be separated by simple distillation. Thus, compared with alkaline systems, our process (i) avoids disproportionation-induced loss of formaldehyde, and (ii) simplifies purification by eliminating the need for an additional acidification step and minimizing product mixing.

We acknowledge that the side reaction (HER) in acidic solution inevitably decrease overall efficiency. Nevertheless, our system already achieves high conversion and good selectivity under the current conditions. Moreover, our preliminary results indicate that tuning the alkyl chains of the CuTAPc layer provides a viable route to suppress HER and enhance the intrinsic activity toward FA reduction.

Supplementary Note Figure 1. **a,c** Disproportionation percentage and corresponding ^1H NMR spectra of 1 M FA + 1 M KOH solution at different time intervals. **b,d** Disproportionation percentage and corresponding ^1H NMR spectra of 0.1 M FA + 1 M KOH solution at different time intervals.

3. The manuscript reports DFT calculations conducted at 0.6 V and pH 1, but the computational methodology provided does not adequately describe how electrode potential and proton activity were explicitly incorporated. Without a clear constant-potential framework or justified approximations, the reliability and physical relevance of these calculations remain questionable.

Reply: We thank the reviewer's suggestion on providing details on computation. We have revised the method part with changes highlighted in red in the manuscript:

Computational tools involved in modeling these systems involved using spin-polarized periodic density functional theory (DFT) as implemented in the Vienna ab initio simulation package (VASP 6.4.2) for geometry optimization and phonon calculations. We used the Perdew-Burke-Emzerhof (PBE) functional with a plane-wave cutoff energy of 500 eV⁹⁴⁻⁹⁹. We used the D3(BJ) empirical van der Waals interactions. For structural optimizations, the Brillouin zone was sampled using a $2 \times 2 \times 1$ K-point grid based on the Gamma scheme. The convergence criteria for forces were set to 0.02 eV/Å, while electronic structure energy convergence was 10^{-6} eV.

To accurately represent the aqueous environment of the electrochemical cell, the model incorporates

the influence of three water layers (24 water molecules) on the thermodynamic stability of adsorbed intermediate species. Solvation can lower the activation barrier for proton transfer steps, especially when formate interacts with water clusters. A trilayer of water mimics the interfacial structure under anodic conditions, where strong hydration and electric double-layer effects dominate.⁹⁹ (<https://pubs.acs.org/doi/10.1021/jp047349j>) Hydrogen bonding interactions with water significantly enhance the stability of surface-bound intermediates.

All DFT calculations were performed at zero applied potential ($U = 0$ V) and at 0 pH within the framework of the computational hydrogen electrode (CHE) model. To account for the effect of applied electrode potential on the free energy of electron transfer steps, a voltage correction was introduced. The electrochemical potential of an electron at potential U is given by: $\mu_e = -eU$; where e is the elementary charge and U is the electrode potential referenced to the CHE. Consequently, for any elementary step involving n_e electrons, the corrected Gibbs free energy is expressed as:

$$\Delta G(U) = \Delta G_{\text{DFT}} - n_e eU$$

Here, ΔG_{DFT} includes zero-point energy (ΔZPE) and entropic corrections ($T\Delta S$) evaluated at $U = 0$ V. This approach effectively shifts the free energy of proton–electron transfer steps to reflect the applied potential, enabling accurate construction of potential-dependent free energy diagrams. Positive potentials stabilize oxidation steps, while negative potentials favor reduction steps. All reported energetics at non-zero potentials were obtained using this correction.

To incorporate pH effects, the free energy of a proton is adjusted by:

$$\Delta G_{\text{pH}} = k_B T \ln(10) \times \text{pH} \approx 0.059 \times \text{pH} \text{ (in eV at 298 K)}$$

Thus, the total free energy correction for a reaction step involving n_e electrons is:

$$\Delta G(U, \text{pH}) = \Delta G_{\text{DFT}} + \Delta ZPE - T\Delta S - n_e eU + n_H \times 0.059 \times \text{pH}$$

This ensures that both applied potential and electrolyte acidity are properly reflected in the computed energetics.¹⁰⁰ (10.1021/acs.jpcclett.4c03408). The oxidation of formaldehyde to formate is thermodynamically favored because positive potential stabilizes electron-removal steps. However, at pH 1, the high proton activity minimizes the destabilizing effect on proton-coupled electron transfer, ensuring that the formation of HCOOH remains energetically accessible under acidic conditions.

4. In TEA, prior studies have demonstrated that noble-metal-free catalysts can achieve competitive FA conversion in alkaline electrolytes, whereas the present work employs precious-metal-based anode catalysts, which substantially increases material costs. The manuscript does not sufficiently justify this choice, nor does it explain how catalyst cost and scalability were rationally accounted for in the TEA, raising concerns about the economic realism of the analysis.

Reply: We thank the reviewer for the valuable comment on the TEA discussion. The use of noble-metal-free catalyst for acidic formaldehyde oxidation has been evaluated in Supplementary Note 2. MnO_2 demonstrates a FE_{HCOOH} approaching 70% in the range of 1.5-1.75 V (Fig. SN3). Under the strongly acidic, oxidizing conditions, MnO_2 presents inferior activity compared with Pt_2Ru . The high intrinsic activity and chemical stability of Pt_2Ru catalyst under these conditions allow operation with an extended replacement interval, thereby reducing the annualized catalyst cost. In addition, we assume recovery and recycling of Pt_2Ru at end of life, consistent with current industrial practice, which further mitigates the effective material cost.

We fully recognize the importance of reducing dependence on precious metals. As indicated by our

MnO₂ results, noble-metal-free catalysts are a promising direction, and future work will focus on improving their stability and activity in acidic media so that they can replace or substantially dilute Pt₂Ru while maintaining performance.

Reviewer #2 (Remarks to the Author):

Authors have well addressed my concerns. I have no additional questions.

Reply: We sincerely appreciate the reviewer's constructive suggestions, which have helped improve the quality of our manuscript. We are also grateful for the reviewer's recognition of our efforts and for the valuable time and support provided throughout the review process.

Reviewer #3 (Remarks to the Author):

The author has satisfactorily addressed the issues raised by the reviewers. It is recommended that the manuscript be accepted.

Reply: We sincerely appreciate the reviewer's constructive suggestions, which have helped improve the quality of our manuscript. We are also grateful for the reviewer's recognition of our efforts and for the valuable time and support provided throughout the review process.